# Children’s Involvement in Different Sport Types Differentiates Their Motor Competence but Not Their Executive Functions

**DOI:** 10.3390/ijerph19095646

**Published:** 2022-05-06

**Authors:** Martha Spanou, Nektarios Stavrou, Aspasia Dania, Fotini Venetsanou

**Affiliations:** School of Physical Education and Sport Science, National and Kapodistrian University of Athens, 17237 Athens, Greece; mspano@phed.uoa.gr (M.S.); nstavrou@phed.uoa.gr (N.S.); adania@phed.uoa.gr (A.D.)

**Keywords:** motor skills, inhibitory control, working memory, cognitive flexibility

## Abstract

Sports provide a context where important aspects of children’s health, such as motor skills and cognitive functions, can be enhanced. However, it is unknown which type of sport may be better for the development of motor competence (MC) and executive functions (EFs). This study investigated potential differences in MC and EFs in boys and girls, being involved in different types of sports (team, individual open skill, individual closed skill). A total of 115 children (49 boys), 8–12 years old (10.30 ± 1.19 years), participated in the study. Their MC was assessed with the Bruininks–Oseretsky Test of Motor Proficiency-2 Short Form, whereas for EFs, the Attention Network Test, the digits backwards test, and the how many–what number test were utilized. Significant MC differences among participants in different types of sports were revealed, favoring those from closed-skill sports; nevertheless, their EFs were at similar levels. Furthermore, no significant gender MC and EFs differences were detected. It seems that children’s participation in specific types of sports differentiates their motor skills but not their EFs, whereas boys and girls, when provided with the same opportunities, present similar levels of MC and EFs.

## 1. Introduction

Executive functions (EFs) are higher-order cognitive processes that are considered necessary for goal-directed behavior [1]. Core EFs are inhibitory control (i.e., the ability to hold on thoughts and impulses [2]), cognitive flexibility (i.e., the ability to switch between two or more cognitive states [3]), and working memory (i.e., the ability to store, process, and use information, when is necessary [4]). Due to the importance of EFs for school readiness [5], academic achievement [6], and mental health [7], a growing body of literature has increasingly focused on environmental factors that may improve (or inhibit) the development of those significant cognitive processes [5,8].

Participation in organized physical activity, such as sports, is thought to significantly contribute to the enhancement of EFs [9]. Several researchers claim that cognitively demanding physical activity and complex motor tasks enhance EFs in children [1,10,11]. These characteristics of coordinatively demanding and cognitively challenging tasks are typical in open-skill sports [12], such as volleyball, tennis, and football, which are characterized by externally-paced acts and are identified by an unpredictable environment that requires adapting to multiple external stimuli [13]. In contrast, closed-skill sports (such as track and field and gymnastics) are characterized by self-paced acts and a relatively predictable training environment [13], with predetermined and consistent responses in terms of motor performance [12]. That is why it is expected that participants in open-skill sports will have higher scores in tests assessing EFs than participants in closed-skill sports.

Indeed, in adult populations, it has been revealed that athletes of open-skill sports have better EFs scores than athletes of closed-skill sports [14,15,16,17]; furthermore, athletes in team sports present higher levels of EFs, compared to those of individual open-skill sports [18,19]. Only Jacobson and Matthaeus [20] have found that closed-skill sports athletes have better inhibitory control than open-skill sports athletes.

Nevertheless, in children there are limited studies that provide conflicting findings. Specifically, some researchers report better performance in EFs tests for children taking part in open-skill sports, with those participating in team sports outperforming their peers from closed-skill sports [21,22]. On the contrary, other researchers have concluded that children involved in different types of sports present similar levels of EFs [12,23].

Another parameter that has been found to be positively affected by children’s participation in organized physical activity is motor competence (MC) [24]. MC refers to an individual’s proficiency in executing a wide range of gross and fine motor skills [25] and is important not only for sport participation [26], but also for the adoption of an active lifestyle [27,28,29]. Gross MC is usually referred to as the proficiency in a range of fundamental motor skills [30] categorized into locomotor, object control, and stability skills [30]. On the other hand, fine motor skills are based on the co-ordination of small muscle groups [30]. Children’s MC is improved with age (suggestively [31,32,33,34,35]); however, developmentally appropriate forms of physical activity are prerequisite for the attainment of optimal MC levels [36,37,38].

In recent years, many children have participated in sports; however, there are only a few studies that compare MC of children from different types of sports. For example, Formenti et al. [12] found that children involved in open-skill sports had better speed and agility scores than those taking part in closed-skill sports, whereas their performance in power and balance tasks was similar. In a relevant study, Mehamad et al. [39] found that children participating in individual open- (badminton, taekwondo) and closed-skill sports (athletics) performed better in locomotor skills, whereas those participating in team sports (basketball, handball, hockey) presented better manipulative skills and had a higher gross motor development as measured by the Test of Gross Motor Development-2 (TGMD-2; [40]).

Summarizing the aforementioned, MC and EFs seem very important for children’s health and quality of life. Sports, as an aspect of organized physical activity, provide a context where MC and EFs can be improved. However, the need to motor adapt in response to the continuously changing environmental constraints becomes less imperative as we move from team sports to individual open-skill sports [19] and individual closed-skill sports [19]. Thus, we may infer that different types of sports may differentially assist MC and EFs enhancement. Nevertheless, until now, only one published study [12] has examined how the type of sport may be associated with children’s MC and EFs.

Therefore, the main purpose of the present study was to examine potential MC and EFs differences in children 8–12 years old being involved in different types of sports (team, individual open skill, individual closed skill). Its secondary aim was to investigate possible gender differences in MC and EFs. In order to obtain a clear picture of the investigated associations, the potential impact of children’s age and years of sport participation was examined and controlled, taken into account that significant relationships between age and MC [31,32,33,34,35], sports participation and EFs [41], as well as age and EFs have been reported [21,42,43,44,45].

## 2. Materials and Methods

### 2.1. Participants

A total of 115 children (*n* = 49 boys, *n* = 66 girls), aged 8–12 years old, were recruited from sports clubs in Athens, Greece, and voluntarily took part in the study. Among them, 43 (*n* = 20 boys, *n* = 23 girls) were involved in team sports (volleyball, football), 33 (*n* = 18 boys, *n* = 15 girls) in individual open-skill sports (tennis), and 39 (*n* = 11 boys, *n* = 28 girls) in individual closed-skill sports (gymnastics, track and field). Prior to the beginning of the study, the children and their parents or legal caretakers were informed about the aim of the study and the procedures that will be followed, and they were asked to give their verbal and written consent, respectively. The study was approved by the Ethics committee of the School of Physical Education and Sport Science, National and Kapodistrian University of Athens.

### 2.2. Measures

#### 2.2.1. Motor Competence

For the assessment of participants’ MC, the Short Form of the Bruininks–Oseretsky Test of Motor Proficiency—Second Edition (BOT-2SF; [46]) was used. The BOT-2SF is a widely used tool for the assessment of gross and fine motor skills in children and youth aged 4–21 years old [46]. It consists of 14 items (drawing lines through crooked paths; folding paper; copying a square; copying a star; transferring pennies; jumping in place with same sides synchronized; tapping feet and fingers with same sides synchronized; walking forward on a line; standing on one leg on a balance beam with eyes open; one-legged stationary hop; dropping and catching a ball; dribbling a ball; knee push-ups; sit-ups) and its total administration time ranges between 15 and 20 min per child.

Each participant’s performance is given as a total point score that can be converted to standard score, percentile rank, or descriptive category, according to the norms provided in the BOT-2SF manual. In the present study, the total point score, as well as the 14 individual items point scores, were used. The technical adequacy of the BOT-2SF is sufficiently supported [46,47]. Moreover, before its utilization in Greece, the psychometric properties were examined, indicating acceptable validity and reliability indices [48].

#### 2.2.2. Executive Functions

The tasks for inhibitory control and cognitive flexibility were administered with E-Prime Software 2.0 (Psychology Software Tools, Pittsburgh, PA, USA), while the task for working memory with Power Point. These tasks were chosen in order to assess the three core components of EFs [49].

For the assessment of inhibitory control, the Attention Network Test (ANT [50,51]), adapted for children into Greek language [52], was used. The design of ANT is based on the Flanker task [53] and includes five arrows as the stimulus. Participants are required to respond based on the direction of the central arrow by pressing the left or the right key on the mouse regardless of the direction of the other four arrows (two arrows on the left and two on the right of the central one). All of the arrows are in a row in the middle of a white screen and in congruent trials are facing the same direction, while in the incongruent trials the central arrow is facing in the opposite direction from the other four arrows. The task consists of four blocks of congruent and incongruent trials with a brief and encouraging break between each block, while in the beginning children, have some practice trials with feedback. If needed, practice trials can be repeated in order for the task to be clear for the participants. Examinee’s performance on the test can be given as reaction time and accuracy in congruent and incongruent trials. In the present study, reaction time and accuracy in congruent and incongruent trials were applied in data analysis. Moreover, an index of inhibitory control was calculated by subtracting the mean reaction time of the correct congruent trials from the mean reaction time of the correct incongruent trials. The psychometric properties of the initial ANT [51,54] and the Greek adaptation [55,56] are sufficiently supported.

Cognitive flexibility was assessed with the task how many–what number [57], which has been previously used in Greek children [58]. The how many–what number has been widely used in studies with children [59,60,61] to assess their ability to differentiate their response according to the rules. The task includes four types of stimuli. Specifically, four different cards consisting of one (1, 3) or three (1 1 1, 3 3 3) digits are presented to the participants. It consists of two blocks, a simple one with non-switch tasks and a complex one which includes switch and non-switch tasks. The simple block consists of two parts. In the first part, children are required to respond to the rule “what number”, identifying the numeric value of the digits presented on the screen and pressing 1 or 3 on the keyboard. In the second one, children respond to the rule “how many” identifying the number of the digits presented on the screen and pressing 1 or 3 on the keyboard, respectively. Each part consists of four practice trials and 24 main ones. In the complex block there are two parts, where children are required to change between these two rules after every second trial. The complex block starts with eight practice trials, followed by the two parts with 36 main trials each. In the present study, accuracy and reaction time in switch trials of complex blocks were used for the assessment of cognitive flexibility. Additionally, switch costs, which arise by the subtraction of mean reaction time of non-switch trials from the mean reaction time of switch trials, were calculated as well.

Finally, for working memory, the task digits backwards, from the Working Memory Test Battery for Children [62] was utilized. This test assesses the ability of the participant to recall a sequence of numbers in reverse order. The sequence is presented at the rate of one digit per second. The test ends when participants fail to repeat three sequences of digits of the same length. The dependent variable for the working memory was the number of correct responses. Several research findings support the technical adequacy of both the original digits backwards [63,64] and its Greek version [42,65,66].

#### 2.2.3. Procedure

Measurements were conducted by the first author, during two meetings at the facilities of each sport club before their training, according to the assessment protocols guidelines. At the first meeting, (a) children’s age and years of sports participation were collected, and (b) the BOT-2SF was administered to each child individually in a properly designed place. The duration of the first meeting was about 25 min per child. In the second meeting, EFs assessment protocols were administered. Each participant sat in front of a laptop and completed the three computer-based tasks for the assessment of their EFs, with a total duration of 20–30 min per participant. The EFs tests were presented in a balanced order to avoid the effect of sequence.

#### 2.2.4. Data Analysis

The data collected were analyzed with the IBM SPSS 25 software package. At a preliminary level, Pearson correlation coefficients were calculated to examine potential associations among children’s age and years of sport participation with their MC and EFs measures. The correlation analysis revealed that participants’ age was statistically significantly correlated with both MC (*p* < 0.001) and EFs measures (*p* < 0.04), whereas years of sport participation was significantly correlated to inhibitory control (*p* < 0.001) and cognitive flexibility (*p* < 0.001) measures. Because of the significant correlations that were revealed, age and years of sport participation were controlled to reduce their potential impact on the analyses.

To examine the potential MC and EFs differences that were associated with participants’ gender and the type of sport, analyses of covariance (ANCOVAs) were implemented on (a) the total BOT-2SF point score and (b) the correct answers on working memory. Moreover, multiple analyses of covariance (MANCOVAs) were performed on (a) the inhibitory control and cognitive flexibility data and (b) the point scores of the 14 individual BOT-2SF items, in order to have a closer look at children’s motor skills. According to the results of the correlation analysis, age was used as a covariate in all the above analyses, whereas “years of sport participation” was set as a covariate only in the analysis of inhibitory control and cognitive flexibility. Post hoc comparisons were conducted with the Sidak test. Alpha was set at 0.05, whereas effect sizes were also taken into consideration, using the η^2^_p_ value for the interpretation of the results. According to Cohen [67], differences of η^2^ values higher than 0.14 are considered of practical importance.

## 3. Results

Descriptive statistics (M, SD) of participants’ age and years of sport participation by type of sport and gender are presented in Table 1. Moreover, estimated marginal means and standard errors of children’s BOT-2SF and EFs scores by type of sport and gender are displayed in Table 2.

Two-way ANCOVA was conducted on the total point score of BOT-2SF, revealing that age was a significant covariate (F = 17.63, *p* < 0.001, η^2^_p_ = 0.14), with older children presenting higher scores. The interaction of gender and type of sport was not significant, whereas boys and girls showed similar MC (Table 2). However, MC was significantly different among the three types of sports. Specifically, children involved in closed-skill sports had significantly higher total point scores than those in the team sports (mean difference = 5.19, *p* < 0.001) and open-skill sports (mean difference = 2.51, *p* = 0.046). Moreover, the participants in open-skill sports had significantly higher total point scores than those in team sports (mean difference = 2.68, *p* = 0.02).

Regarding the 14 BOT-2SF item point scores, the MANCOVA applied showed that age was a significant covariate (Pillai’s trace = 0.36, F (13,96) = 4.19, *p* < 0.001, η^2^_p_ = 0.36). The interaction of gender and type of sport (Pillai’s trace = 0.17, F (26,194) = 0.69, *p* = 0.87, η^2^_p_ = 0.09) did not significantly differentiate the results. In contrast, both gender (Pillai’s trace = 0.24, F (13,96) = 2.31, *p* = 0.01, η^2^_p_ = 0.24) and type of sport revealed significant MC differences (Pillai’s trace = 0.51, F (26,194) = 2.54, *p* < 0.001, η^2^_p_ = 0.25).

Univariate analyses that followed (Table 2) showed statistically significant differences between boys and girls in folding paper, transferring pennies, favoring girls, and dribbling a ball, favoring boys. Differences among the types of sports emerged in folding paper, tapping feet and fingers with same sides synchronized, one-legged stationary hop, knee push-ups, and sit-ups, as is presented in Table 2. Nevertheless, as it can be noticed, among the above differences, only those in sit-ups among participants in different types of sports were of practical importance. In actuality, children taking part in individual closed-skill sports achieved significantly higher point scores than those in individual open-skill sports (mean difference = 0.67, *p* = 0.02) and team sports (mean difference = 1.43, *p* < 0.001). Moreover, participants in individual open-skill sports presented higher point scores than those participated in team sports (mean difference = 0.77, *p* = 0.003).

Regarding EFs, the ANCOVA applied for the working memory revealed that age was a statistically significant covariate, though of no practical importance (F = 5.74, *p* = 0.02, η^2^_p_ = 0.06). Neither the interaction between gender and type of sport nor gender presented significant main effects (Table 2). The main effect of type of sport was statistically significant, though of no practical importance, with participants in open-skill sports showing higher performance in the digits backward task compared to team sports participants (mean difference = 2.52, *p* = 0.04).

As far as inhibitory control and cognitive flexibility are concerned, according to the MANCOVAs, age was a significant covariate for both inhibitory control (Pillai’s trace = 0.29, F (6,92) = 6.34, *p* < 0.001, η^2^_p_ = 0.29) and cognitive flexibility (Pillai’s trace = 0.23, F (3,95) = 9.66, *p* < 0.001, η^2^_p_ = 0.23), indicating that older children performed better than the younger ones. Conversely, “years of sports participation” was not a significant covariate for either the inhibitory control (Pillai’s trace = 0.08, F (6,92) = 1.24, *p* = 0.29, η^2^_p_ = 0.08) or cognitive flexibility (Pillai’s trace = 0.03, F (3,95) = 1.08, *p* = 0.36, η^2^_p_ = 0.03). Regarding the independent variables, the only significant effect that appeared was that of the type of sport in cognitive flexibility (Pillai’s trace = 0.13, F (6,192) = 2.16, *p* = 0.05, η^2^_p_ = 0.06); however, η^2^_p_ did not exceed the value of 0.14. Sidak multiple comparisons did not reveal any significant differences among the participants of the three types of sports.

## 4. Discussion

Taking into account that participation in sports is associated with MC and EFs, two important factors for health and quality of life, the purpose of this study was to examine possible MC and EFs differences in children 8–12 years old, who are involved in different types of sports (team, individual open skill, individual closed skill). Moreover, possible MC and EFs gender differences were investigated, whereas the effects of children’s age and years of sport participation was examined and controlled. Our key finding was that children’s involvement in different types of sports differentiated their MC, but not their EFs. Moreover, there were not significant MC and EFs gender differences, whereas age (but not years of sport participation) was strongly associated with both factors.

Starting with the comparison among different types of sports, it was revealed that children taking part in individual closed-skill sports (gymnastics, track and field) had the highest total BOT-2SF point scores, followed by their peers participating in individual open-skill sports (tennis) and those from team sports (volleyball, football), who presented the lowest scores. This is in contrast to the findings of the study of Mehamad et al. [39], in which children participating in team sports achieved the highest TGMD [40] scores (although those taking part in individual open- and closed-skill sports performed better in the locomotor items of the TGMD). This discrepancy can be attributed to the different MC assessment tools that were implemented. In the study of Mehamad et al. [39], the TGMD [40] that focuses only on gross motor skills was used; in our study, the BOT-2SF was utilized in order to obtain a comprehensive picture of MC. It is well known that different motor assessment tools shed light into different aspects of motor development [68]. However, if an authentic measure of MC had been used both in our study and that of Mehamad et al. [39], a different picture of MC would have been obtained. Another explanation for the higher MC levels of children participating in closed-skill sports in this study may be the fact that the training of both gymnastics and track and field focuses on the development of fundamental motor skills such as locomotor and stability skills, which are highly connected with the athletic skills of these sports. When children’s performance in the 14 individual items of the BOT-2SF was examined, the only practically important difference was spotted in sit-ups, in favor of closed-skill sports participants. One possible explanation for this finding could be that in gymnastics, the training focuses on bodyweight exercises for the increase of children’s relative strength. This notion was supported by Sheerin et al. [69], who showed that children’s score on the 30 s sit-up test was improved significantly after a 9-week gymnastics program. In contrast, in the training of team sports, such as volleyball and football, as well as of individual open-skill sports, such as tennis, more emphasis is given to technique and game-based activities.

Apart from the MC comparisons among the different types of sports, the “good news” is that the participants presented, on average, high levels of MC. Indeed, a closer look in Table 2 reveals that the average total BOT-SF2 point score in each group was 70 points and above; these scores are high, taking into account that the BOT-2SF assesses MC in children and youth aged 4–21 years and its point score range is between 0 and 88 points. This is very important, since a sufficient level of MC is thought to be connected with an active lifestyle [70] and higher levels of cognitive functions [49].

Moreover, in the present study, no EFs differences were detected among children who take part in different types of sports. This is in contrast to a number of studies in adults, in which open-skill sports athletes were found to have higher levels of EFs than athletes of closed-skill sports [14,15,16,17]. Nevertheless, our findings are in line with studies with children who participated in similar sports. Russo et al. [23] identified a lack of differences in working memory between participants in open- (tennis, basket, football) and closed-skill sports (track and field, swimming, rhythmic gymnastics) in preadolescents. Furthermore, Formenti et al. [12] were led to the same conclusion, comparing inhibitory control between participants in football, basketball, volleyball, and martial arts and those in swimming, rhythmic gymnastics, and ballet, where they only detected better accuracy in incongruent trials in favor of open-skill participants. However, in other studies, significant EFs differences among children participating in different types of sports are reported. For example, De Waelle et al. [21], in a sample of 170 girls aged 8–12 years, figured out that team sports participants significantly outperformed those of closed-skill sports in EFs. However, it should be noticed that De Waelle et al. [21], despite assessing children’s EFs with seven items from Cambridge Brain Sciences (CBS) test battery, used only one weighted score derived from those items for data analysis. This might have differentiated their results, impeding a comprehensive examination of potential differences between children. Indeed, a closer look at the descriptive statistics table of their study supports the notion that closed-skill sports participants had higher scores than those of open-skill sports in three CBS items, which may have been masked in the analyses. Moreover, in the study of Ludyga et al. [22], with slightly older participants (n = 184, 9–14 years old), a positive association emerged only between inhibitory control and open-skill sports participation and not with closed-skill sports participation. The lack of differences in EFs revealed in the current study could be attributed to the similarity of sport training content in childhood. It is known that in beginners, training in different sports is characterized by common elements which are relatively independent of the type of sport [23]. Moreover, in the majority of sports (either team, closed, or open skill), competitive relays constitute a common activity in children groups [23].

As far as the role of gender is concerned, our results revealed that there were no significant gender differences in either MC or EFs. Starting with MC, our findings are in agreement with previous studies in Greek children of the same age. Both Afthentopoulou et al. [71], using the BOT-2SF [46], and Kaioglou et al. [34], using the CAMSA [72], report that there are no practically significant MC differences between boys and girls. Focusing on the 14 items of BOT-2SF, statistically significant (yet of no practical importance) differences came out that can be attributed to social factors, which, up to a point, form the roles of boys and girls in the society [25,73], leading boys to engage with ball games and girls with activities such as dancing and gymnastics [73], taking into account that at this age, children do not differ biologically [25]. Although of no practical importance, those MC differences should be considered for the training programs improvement in order to provide children with the necessary opportunities for the development of their MC. Regarding EFs, boys and girls presented similar scores; a finding that is in agreement with the study of Russo et al. [23], in which the two genders had similar EFs scores. Furthermore, our results are supported by several meta-analyses, which provide evidence about the similar performance of boys and girls in EFs [74,75,76].

Finally, as was expected, age was strongly associated with children’s MC, a finding which is in agreement with a large number of relevant studies (suggestively [31,32,33,34,35]. Similar was the pattern in EFs, a finding that is evident in studies examining EFs in children [21,44,45], and this can be attributed to the development of the prefrontal cortex [45] that occurs across age.

This study has some limitations that should be taken into account when interpreting its results. First of all, the classification of sports used in this study (and in several previous ones) masks slight differences between the sports classified into the same category. It is well known that purely closed and purely open skills represent the two edges of a continuum that includes several degrees of environment predictability [77]. Thus, volleyball, for example, is “less open” than football. In addition, the cross-sectional design of the study does not allow for conclusions of cause–effect relationships between children’s sports participation and their MC and EFs levels. Moreover, participants were recruited only from a big city of Greece, Athens; thus, other factors relating to their daily life may have affected the results. However, this is the first study examining possible differences in the three EFs and MC in children who take part in different type of sports, taking into account confounding factors, such as children’s age and years of their sport participation.

## 5. Conclusions

It can be concluded that school-aged children who participate in team, individual open skill-, and individual closed-skill sports differ significantly in their MC, but not in EFs. Moreover, older children perform better in MC and EFs, whereas boys and girls present similar MC and EFs levels. Although this study provides valuable information regarding this topic, research evidence is limited so far. Further research is needed in order to explore the impact of other factors, such as socioeconomic status, sedentary behavior, sleep and screen time, on both MC and EFs. Furthermore, longitudinal studies investigating the development of MC and EFs in children taking part in different types of sports compared to their physically inactive peers will provide fruitful information with regard to the potential of sports to contribute to children’s health and development.

## Figures and Tables

**Table 1 ijerph-19-05646-t001:** Means (M) and standard deviations (SD) of participants’ age and years of sport participation by sport type and gender.

	Boys			Girls		
	Team	OS	CS	Team	OS	CS
Age	10.65 ± 1.36	10.61 ± 0.83	10.72 ± 1.43	10.06 ± 0.98	10.05 ± 1.11	10.03 ± 1.30
Years of Sport Participation	4.25 ± 2.32	4.23 ± 2.39	2.73 ± 1.85	1.8 ± 1.17	3.36 ± 1.34	3.36 ± 4.88

OS: open-skill sports, CS: closed-skill sports.

**Table 2 ijerph-19-05646-t002:** Estimated marginal means (+ standard errors) and ANCOVA results for BOT-2SF and EF scores by gender and type of sport.

		Boys			Girls			ANCOVA Results		
		Team	OS	CS	Team	OS	CS	Gender × ToS	Gender	ToS
BOT-2SF	TPS	70.67 ± 0.92	72.58 ± 0.97	75.52 ± 1.24	70.03 ± 0.86	73.48 ± 1.06	75.56 ± 0.78	F = 0.33, η^2^_p_ = 0.01	F = 0.02, η^2^_p_ = 0.00	F = 14.79 **, η^2^_p_ = 0.22
	Drawing lines through paths	6.85 ± 0.08	7 ± 0.08	6.82 ± 0.11	7 ± 0.07	7 ± 0.09	7 ± 0.07	F = 0.51, η^2^_p_ = 0.01	F = 1.84, η^2^_p_ = 0.02	F = 0.83, η^2^_p_ = 0.02
	Folding paper	5.02 ± 0.32	5.47 ± 0.33	6.33 ± 0.42	6.11 ± 0.29	6.16 ± 0.36	6.67 ± 0.27	F = 0.66, η^2^_p_ = 0.01	F = 6.26 *, η^2^_p_ = 0.06	F = 4.18 *, η^2^_p_ = 0.07
	Copying a square	4.90 ± 0.06	4.9 ± 0.06	4.91 ± 0.08	4.83 ± 0.06	5 ± 0.07	5 ± 0.05	F = 1.38, η^2^_p_ = 0.03	F = 0.83, η^2^_p_ = 0.01	F = 1.34, η^2^_p_ = 0.02
	Copying a star	4 ± 0.15	4.06 ± 0.16	3.93 ± 0.20	3.95 ± 0.14	4.11 ± 0.17	4.26 ± 0.13	F = 0.75, η^2^_p_ = 0.01	F = 0.68, η^2^_p_ = 0.01	F = 0.41, η^2^_p_ = 0.01
	Transferring pennies	5.82 ± 0.21	5.94 ± 0.22	6.30 ± 0.28	6.35 ± 0.20	6.69 ± 0.24	6.81 ± 0.18	F = 0.17, η^2^_p_ = 0.00	F = 10.08 *, η^2^_p_ = 0.09	F = 2.34, η^2^_p_ = 0.04
	Jumping in place	3 ± 0.05	2.94 ± 0.05	3 ± 0.06	3.09 ± 0.04	3 ± 0.05	3 ± 0.04	F = 0.38, η^2^_p_ = 0.01	F = 1.61, η^2^_p_ = 0.02	F = 1.10, η^2^_p_ = 0.02
	Tapping feet and fingers	3.89 ± 0.05	3.99 ± 0.05	3.99 ± 0.07	3.88 ± 0.05	4 ± 0.06	3.97 ± 0.04	F = 0.05, η^2^_p_ = 0.00	F = 0.00, η^2^_p_ = 0.00	F = 3.10 *, η^2^_p_ = 0.05
	Walking forward on a line	4 ± 0	4 ± 0	4 ± 0	4 ± 0	4 ± 0	4 ± 0	-	-	-
	Standing on one leg on a balance beam	3.76 ± 0.13	3.73 ± 0.14	4.01 ± 0.18	3.82 ± 0.12	3.6 ± 0.15	3.92 ± 0.11	F = 0.31, η^2^_p_ = 0.01	F = 0.27, η^2^_p_ = 0.02	F = 2.26, η^2^_p_ = 0.04
	One-legged stationary hop	8.33 ± 0.17	8.76 ± 0.17	8.98 ± 0.22	8.32 ± 0.15	8.68 ± 0.19	8.59 ± 0.14	F = 0.65, η^2^_p_ = 0.01	F = 1.18, η^2^_p_ = 0.00	F = 4.39 *, η^2^_p_ = 0.08
	Dropping and catching a ball	4.95 ± 0.09	4.94 ± 0.10	5 ± 0.12	4.87 ± 0.09	4.94 ± 0.11	4.79 ± 0.08	F = 0.50, η^2^_p_ = 0.00	F = 1.35, η^2^_p_ = 0.01	F = 0.11 *, η^2^_p_ = 0.00
	Dribbling a ball	6.35 ± 0.33	6.75 ± 0.34	6.25 ± 0.44	4.98 ± 0.30	6.07 ± 0.38	5.79 ± 0.28	F = 1.03, η^2^_p_ = 0.02	F = 8.19*, η^2^_p_ = 0.07	F = 2.43, η^2^_p_ = 0.04
	Knee push-ups	4.21 ± 0.29	4.18 ± 0.31	5.04 ± 0.40	3.55 ± 0.27	3.83 ± 0.34	4.64 ± 0.25	F = 0.15, η^2^_p_ = 0.00	F = 3.16, η^2^_p_ = 0.03	F = 5.44 *, η^2^_p_ = 0.09
	Sit-ups	5.53 ± 0.22	5.95 ± 0.23	6.85 ± 0.29	5.3 ± 0.20	6.42 ± 0.25	6.85 ± 0.18	F = 1.23, η^2^_p_ = 0.02	F = 0.17, η^2^_p_ = 0.00	F = 20.32 **, η^2^_p_ = 0.27
EFs protocols	WM	12.68 ± 0.97	14.61 ± 1.01	14.40 ± 1.32	13.41 ± 0.87	16.52 ± 1.16	14.98 ± 0.87	F = 0.23, η^2^_p_ = 0.01	F = 1.50, η^2^_p_ = 0.02	F = 3.34 *, η^2^_p_ = 0.06
	Interference control	97.94 ± 17.04	109.47 ± 16.87	90.76 ± 21.36	114.44 ± 15.01	99.16 ± 18.72	76.83 ± 14.36	F = 0.50, η^2^_p_ = 0.01	F = 0.29, η^2^_p_ = 0.00	F = 0.98, η^2^_p_ = 0.02
	Switch costs	57.54 ± 93.46	63.94 ± 92.50	94.27 ± 117.11	14.35 ± 82.31	240.67 ± 102.67	−16.68 ± 78.72	F = 1.23, η^2^_p_ = 0.03	F = 0.01, η^2^_p_ = 0.00	F = 0.98, η^2^_p_ = 0.02

OS: open-skill sports, CS: closed-skill sports, ToS: type of sport, TPS: total point score on BOT-2SF, WM: correct answers in the task digits backwards for working memory, * *p* < 0.05, ** *p* < 0.001.

## Data Availability

The datasets generated during and/or analyzed during the current study are available from the corresponding author upon reasonable request.

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
