# Peer review of "Children’s Involvement in Different Sport Types Differentiates Their Motor Competence but Not Their Executive Functions"

_ijerph, 2022, doi:10.3390/ijerph19095646_

Round 1
Reviewer 1 Report
Congratulations to the authors. The article is very interesting and well written. The methodology applied is correct, although some slight improvements are suggested. The introduction and discussion is based on numerous, powerful and up-to-date studies.
The following improvements are recommended:
The classification of types of sports needs more theoretical argumentation. It is based on a single article which also does not show any theoretical references. There are other classifications, for example the classical Parlebas classification, which has been widely used. The authors should argue this choice further.
The results are not well understood. Table 2 should show the comparisons between groups. It would be clearer to make more tables and not just write the results in writing, it is very difficult to understand.
Author Response
Dear reviewer,
First, we would like to thank you for your encouraging feedback regarding the quality of our work and your comments towards the improvement of the manuscript. We attempted to address all the comments in the most appropriate way. Our response to each separate comment are given below
Congratulations to the authors. The article is very interesting and well written. The methodology applied is correct, although some slight improvements are suggested. The introduction and discussion is based on numerous, powerful and up-to-date studies.
The following improvements are recommended:
1)The classification of types of sports needs more theoretical argumentation. It is based on a single article which also does not show any theoretical references. There are other classifications, for example the classical Parlebas classification, which has been widely used. The authors should argue this choice further.
-Thank you very much for noticing the “poor” reference we have used regarding the classification of types of sports. We have changed it with the reference of Singer (2000), who provides a sound theoretical basis. Although the classification of Parlebas is very interesting, the most recent studies in children of this age use the classification of open and closed skill sports is used [suggestively: De Waelle, S., Laureys, F., Lenoir, M., Bennett, S. J., & Deconinck, F. J. (2021); Formenti, D., Trecroci, A., Duca, M., Cavaggioni, L., D’Angelo, F., Passi, A., ... & Alberti, G. (2021); Holfelder, B., Klotzbier, T. J., Eisele, M., & Schott, N. (2020)]; thus, we prefer following this one. Taking into account that (a) the other four reviewers of this paper congratulated us for the well written introduction and (b) in the introduction a clear picture is provided regarding what a closed- or an open-skill sport is, if you agree, we would like to keep the introduction in its current form.
2)The results are not well understood. Table 2 should show the comparisons between groups. It would be clearer to make more tables and not just write the results in writing, it is very difficult to understand.
-Thank you very much for your recommendation. We have added columns with the results of the ANCOVAs regarding the comparisons between groups in Τable 2; thus, only the results of multivariate analyses are reported in detail in text. We hope that now the results are clearer.
Reviewer 2 Report
This study investigates potential differences in MC and EF in boys and girls, practicing different types of sports in Athens. This is an interesting and valuable study, and overall a solid manuscript.
A few minor comments and points for individual sections are offered below.
Introduction
- Well written introduction summarising the main arguments and findings within the extant literature. One thing that would be nice, though, is if you briefly defined fine/gross motor skills
Results
- At line 243, what do you mean by 'no practical importance'?
Discussion/Conclusion
- I would encourage you to discuss some potential explanatory reasons as to why individual, closed skill sports had the highest MC as per the BOT-2SF. As you note, this is a contrast to other studies. Is it really only because of the different measurements? And, if so, why? Or can it be that the closed skill sports you used (gymnastics, track and field) require participant to develop a broader range of Fundamental Movement Skills (FMS) than the other sports? For instance, in Track, athletes regularly practice locomotor or stability-related movement skills like running, skipping, jumping, balancing, turning, twisting, etc.
- Line 306, please correct 'Formenti were leaded' to 'were led'
- In the conclusion, I would also encourage you to be more precise about what you see as valuable potential research directions. What topics, target groups and methodologies do you suggest be used?
- Finally, though the English in the manuscript is quite good, I would suggest the authors undertake a final round of proofreading.
Author Response
Dear reviewer,
First, we would like to thank you for your encouraging feedback regarding the quality of our work and your comments towards the improvement of the manuscript. We attempted to address all the comments in the most appropriate way. Our response to each separate comment are given below
This study investigates potential differences in MC and EF in boys and girls, practicing different types of sports in Athens. This is an interesting and valuable study, and overall a solid manuscript.
A few minor comments and points for individual sections are offered below.
1) Introduction
Well written introduction summarising the main arguments and findings within the extant literature. One thing that would be nice, though, is if you briefly defined fine/gross motor skills
-Thank you for the proposal. Information about fine and gross motor skills was added in the introduction of the manuscript (lines 62-65).
2)Results
At line 243, what do you mean by 'no practical importance'?
-In the results we provide data about p-values which refer to statistical significance and effect sizes (η2), which refer to practical significance/ importance. With the phrase “no practical importance” we mean that the effect of the independent variable is not large enough to be meaningful in the real world.
3)Discussion/Conclusion
I would encourage you to discuss some potential explanatory reasons as to why individual, closed skill sports had the highest MC as per the BOT-2SF. As you note, this is a contrast to other studies. Is it really only because of the different measurements? And, if so, why? Or can it be that the closed skill sports you used (gymnastics, track and field) require participant to develop a broader range of Fundamental Movement Skills (FMS) than the other sports? For instance, in Track, athletes regularly practice locomotor or stability-related movement skills like running, skipping, jumping, balancing, turning, twisting, etc.
-This paragraph has been enriched.
Line 306, please correct 'Formenti were leaded' to 'were led'
-It has been corrected (line 326).
In the conclusion, I would also encourage you to be more precise about what you see as valuable potential research directions. What topics, target groups and methodologies do you suggest be used?
-Thank you very much for your suggestion; however, we believe that clear potential research directions are already provided in the conclusion.
Finally, though the English in the manuscript is quite good, I would suggest the authors undertake a final round of proofreading.
The text has been proofread.
Reviewer 3 Report
dear authors
congratulations for the efforts.
some minor revision are necessary in my view.
- the EF was also assessed considering high level o low level of techincal skills (Percept Mot Skills. 2021 Dec;128(6):2710-2724)
- the tennis is an individual sport but not fully open such as volleyball is not fully open (in point of this Wrisberg & Schmidt defined the PA in three different cluster: predictable, no predctable and semi-predctable)
- the growth is a natural phenomenon that lead to MC improvements (Front Pediatr. 2021 Sep 10;9:738294)
thus, this three point must mentioned in discussion and cited as a limit of the study.
Author Response
Dear reviewer,
First, we would like to thank you for your encouraging feedback regarding the quality of our work and your comments towards the improvement of the manuscript. We attempted to address all the comments in the most appropriate way. Our response to each separate comment are given below
Dear authors congratulations for the efforts. Some minor revision are necessary in my view.
1)the EF was also assessed considering high level vs low level of techincal skills (Percept Mot Skills. 2021 Dec;128(6):2710-2724)
2)the tennis is an individual sport but not fully open such as volleyball is not fully open (in point of this Wrisberg & Schmidt defined the PA in three different cluster: predictable, no predctable and semi-predctable)
3)the growth is a natural phenomenon that lead to MC improvements (Front Pediatr. 2021 Sep 10;9:738294)
thus, this three point must mentioned in discussion and cited as a limit of the study.
-(1)Thank you very much for your proposal. Starting with the study of Lovecchio et al (2021), we totally agree that it is a very interesting paper examining potential executive function-and agility differences between elite and low-division male soccer players. Although the association between the level of athletes and their executive functions is a topic in which several research teams (e.g. Huijgen et al., 2015; Vaughan & Edwards, 2020; Vaughan & Laborde, 2021; Verburgh, Scherder, Lange, & Oosterlaan, 2014) have published relevant papers regarding athletes of various age groups, we prefer examining the association between EF and sport under a developmental prism (i.e., the benefits that that sports can offer to children for their optimal development and health). Talking about 8-12-year old “elite” athletes is not developmental. That is why in this paper we did not focus on that.
-(2) Thank you very much for noticing that. We have taken it into account in limitations.
-(3) Regarding the impact of growth on MC, we totally agree with you. MC is an index of motor development, which in turn is a result of (a) individual’s growth and (b) the interaction of the individual with their environment. Our acknowledgement of the above on children’s MC can be seen in the introduction, where it is reported “Children’s MC is improved with age (suggestively: [29-33]); however, developmentally appropriate forms of physical activity are prerequisite for the attainment of optimal MC levels [34-36]. Moreover, we have taken into account the potential effect of age on the results and we have set it as a covariate in data analyses.
Reviewer 4 Report
Title: Children’s participation in different sports types differentiates their motor competence but not their executive functions
This study investigated potential differences in MC and EF in boys and girls, practicing different types of sports
The following items are suggested for further revision.
- Title: The term “participation” may mislead the reader that this study is going to exam the effect of sports types on motor competence and executive functions
- Line 13: The term “practicing” was not examined in this study. It seems that “selecting” should be used instead.
- Line 82: The 8-12 years old children was selected in the study but the background information about this age group is missing. As age is highly related to MC, 8-12 years old may induce big difference of MC within group. Further literature or supplementary information need to be addressed.
- Line 165: The timeline of first and second meeting didn’t indicate if the data collection was conducted before or after the sports participation. This indication will be important to show if the data could reflect the impact of sports participation or sports selection.
Author Response
Dear reviewer,
First, we would like to thank you for your encouraging feedback regarding the quality of our work and your comments towards the improvement of the manuscript. We attempted to address all the comments in the most appropriate way. Our response to each separate comment are given below
This study investigated potential differences in MC and EF in boys and girls, practicing different types of sports. The following items are suggested for further revision.
1)Title: The term “participation” may mislead the reader that this study is going to exam the effect of sports types on motor competence and executive functions
-The term “participation” was replaced with “involvement”.
2)Line 13: The term “practicing” was not examined in this study. It seems that “selecting” should be used instead.
-Thank you for your proposal. We replaced the word “practicing”; however, selecting might not be the best choice. To start with, selecting does not mean a regular participation/training. Moreover, actually we cannot know if children or their parents have selected the sport in which they are participate…
3)Line 82: The 8-12 years old children was selected in the study but the background information about this age group is missing. As age is highly related to MC, 8-12 years old may induce big difference of MC within group. Further literature or supplementary information need to be addressed.
-We totally agree with you regarding the important impact of age on children’s MC. As you can see earlier in the introduction, it is stated: “Children’s MC is improved with age (suggestively: [29-33])…”. Moreover, when stating the purpose of this study we say: “In order to obtain a clear picture of the investigated associations, the potential impact of children’s age and years of sport participation was examined and controlled, taken into account that significant relationships between age and MC [29-33], sports participation and EF [39], as well as age and EF have been reported [20, 40-43].
Line 165: The timeline of first and second meeting didn’t indicate if the data collection was conducted before or after the sports participation. This indication will be important to show if the data could reflect the impact of sports participation or sports selection.
Data collection was conducted before children’s training. This information has been added (line 176).
Round 2
Reviewer 4 Report
Spell check please such as Line 40